# Legacy Effects of Biochar and Compost Addition on Arbuscular Mycorrhizal Fungal Community and Co-Occurrence Network in Black Soil

**DOI:** 10.3390/microorganisms10112137

**Published:** 2022-10-28

**Authors:** Ying Xin, Yi Fan, Olubukola Oluranti Babalola, Ximei Zhang, Wei Yang

**Affiliations:** 1Institute of Environment and Sustainable Development in Agriculture, Chinese Academy of Agricultural Sciences, Beijing 100081, China; 2Food Security and Safety Focus Area, Faculty of Natural and Agricultural Sciences, North-West University, Mmabatho 2735, South Africa

**Keywords:** biochar, compost, legacy effect, AM fungi, Miseq sequencing, black soil

## Abstract

Compost and biochar are beneficial soil amendments which derived from agricultural waste, and their application was proven to be effective practices for promoting soil fertility. Arbuscular mycorrhizal (AM) fungi form symbiotic associations with most crop plant species, and are recognized as one group of the most important soil microorganisms to increase food security in sustainable agriculture. To understand the legacy effects of compost and biochar addition on AM fungal communities, a field study was conducted on the Songnen Plain, Northeast China. Two years after application, compost addition improved soil aggregate stability, but we did not detect a legacy effect of compost addition on AM fungal community. Our results indicated that AM fungal Shannon diversity and Pielou evenness indices were significantly increased by one-time biochar addition, but unaffected by compost addition after two year’s application. PERMANOVA analysis also revealed a legacy effect of biochar addition on AM fungal community. Network analysis revealed a dramatically simplified AM fungal co-occurrence network and small network size in biochar added soils, demonstrated by their topological properties (e.g., low connectedness and betweenness). However, AM fungal community did not differ among aggregate fractions, as confirmed by the PERMANOVA analysis as well as the fact that only a small number of AM fungal OTUs were shared among aggregate fractions. Consequently, the current study highlights a stronger legacy effect of biochar than compost addition on AM fungi, and have implications for agricultural practices.

## 1. Introduction

Black soil is regarded as one of the most fertile soils distributed in Northeast China [1]. It accounts for approximately 20% of the national arable land and plays a crucial role in ensuring national food security [2]. However, serious soil fertility deterioration has arisen in this region over the past several decades due to extensive agricultural practices [3], which is a huge threat to crop production. To mitigate these problems, effective agricultural practices should be taken in time to improve the quality of black soil.

The application of compost, derived from agricultural waste composting, is considered an effective practice for promoting soil fertility [4]. A wealth of research has shown that compost amendment can increase soil organic matter content [5,6], improve soil quality [7,8] and crop yield [9]. Biochar is another beneficial soil amendment, which is produced through the pyrolysis of agricultural wastes under limited oxygen [10,11]. Biochar addition could increase soil carbon sequestration, improve soil water holding capacity, and reduce nutrient loss due to their huge superficial area and porosity [5,6,7,8,12,13,14].

Even though, constraints still exist regarding the influences of biochar incorporation into agroecosystems [15,16]. One limitation is that biochar would contain some toxic substances (e.g., polycyclic aromatic hydrocarbons) that were produced as by-product from the pyrolysis process [16]. Another limitation is related to its high cost and inconvenience in application and operation [15]. However, compared with compost or other fertilizers, biochar was more recalcitrant due to its higher aromaticity and greater C condensation, and thus could be resident in soil for hundreds of years [17,18]. A long-running field study has indicated that the beneficial effects of biochar addition on soil fertility and crop productivity could be detected after 2–5 years of application [19,20]. Therefore, the legacy effect of biochar addition would make the application of biochar much more convenient because there is no need to apply annually. 

Arbuscular mycorrhizae (AM) are symbiotic associations formed between 80% terrestrial plant roots and soil fungi of the Glomeromycota, and the most prevalent fungi in soil [21,22]. AM fungi can provide benefits to host plants in many ways: (1) act as an extension of roots to increase the soil volume for essential nutrient uptake [23,24]; (2) increase plant tolerance to environmental stress, and induce plant resistance to pathogens [25,26]; (3) develop soil aggregation and improve soil quality [27]. Consequently, AM fungi are receiving growing interests as biofertilizers and are recognized as one of the most important soil microorganisms to increase food security in future sustainable agriculture [28,29,30,31,32].

AM fungi generally benefit from application of compost [33,34,35]. Unlike chemical fertilizers, compost provides a sustained supply of nutrients for AM fungi without adverse impacting on soil pH [33]. Alternatively, it was reported that the humic substances in compost could directly stimulate AM fungal growth [36]. Therefore, previous studies reported that compost addition could enhance AM fungal growth, sporulation and diversity [22,33]. However, the studies that examined the effects of biochar addition on AM fungi yielded divergent results [37,38,39,40]. Some studies provided pieces of evidence that biochar addition would increase AM root colonization [38,39,40], while negative or neutral effects of biochar on AM fungi were also occasionally reported [37,38]. Furthermore, the studies mentioned above mainly focused on AM colonization in roots or soils, while rarely focused on AM fungal community composition and their co-occurrence networks [41].

Most of what we know about AM fungal community was obtained through research at the local, regional or global scale [28,33,42]. However, AM fungal community at the micro-scale such as at the soil aggregate level, is poorly understood [43]. In the current study, we explored the legacy effects of compost and biochar on the AM fungal community and network in different soil aggregate fractions in the soybean agroecosystem. We hypothesized as follows: (1) biochar would exhibit a stronger legacy effect than compost; (2) one-time biochar and compost addition would influence AM fungal community composition and networks; (3) soil aggregate fractions would be a strong determinant for AM fungal community.

## 2. Materials and Methods

### 2.1. Field Description and Environmental Design

This study was conducted at the Xiangyang experimental farm, Harbin, China (45°45’ N, 126°54’ E). This study site has a typical monsoon climate of 4–5.5 °C average annual temperature and precipitation of about 400–500 mm [44]. The soil is typical black soil and has a loamy texture (classified as Mollisols). 

The field trial was established in 2018 with a two-way factorial design (compost addition and biochar addition). There were four treatments: (1) no-biochar addition and no-compost addition (CK); (2) biochar addition without compost addition (B), with biochar application rate of 10 t ha^−1^; (3) compost addition without biochar addition (C), with application rate approximately equal to 180 kg N ha^−1^ (ca. compost 10 t ha^−1^); (4) compost mixed with 10% biochar (BC), with total application rate approximately equal to 180 kg N ha^−1^. Each treatment was repeated four times, resulting in 16 randomly arranged plots (5 m × 5 m each and 2 m separated from each other) totally. The compost used in the present study was obtained through an on-farm composting of cow manure and maize straw. The biochar was produced using rice straw under slow pyrolysis of 450–500 °C and supplied by Sanli New Energy Company (Henan, China). The application rate of compost and biochar was equivalent to the recommended amount of fertilizer in this area. The biochar and compost addition were one-time addition, which was only applied in 2018, and discontinued to applicate in 2019. A detailed description of compost and biochar have been described in Bello et al. [10]. The plots were tilled to a depth of 20 cm before planting and were weeded manually during the growing season. All plots received no chemical fertilizer, herbicide or insecticide.

### 2.2. Soil Sampling, Aggregate Fractionation and Determination of Soil Variables

Soil sampling was collected on 15 September 2019, 1.5 years after one-time application. In detail, three soil cores (10 cm × 10 cm × 10 cm) were carefully collected using a spade and placed in one plastic box for each plot. In total, 16 undisturbed soil samples were collected and transported to the laboratory with no damage. 

The aggregate fractionation procedure was in accordance with the description by Bach and Hofmockel [45]. Briefly, the field-moist soil samples were placed in the freezer(Haier, Qingdao, China) (4 °C) to achieve approximately 10% gravimetric water content. Then 500 g soils were placed on the top of a stack of sterile sieves, with 2 mm-sieve was above the 0.25-mm sieve. The set of sieves was placed onto a sieve shaker (Techang, Xinxiang, China) and shaken at 200 rpms for 2 min. Then each sample was divided into three aggregate fractions: (1) large macroaggregate (>2 mm), (2) little macroaggregate (0.25 mm–2 mm), (3) microaggregate (<0.25 mm). Soils were weighed for each aggregate fraction and the proportion of each fraction was calculated. Therefore, there were 48 samples (4 treatments × 4 replicates × 3 fractions) after aggregate fractionation. Soil organic matter (SOM), total nitrogen (TN), total phosphorus (TP), available potassium (AK), available P (AP), and available N (AN) were determined for each sample, the method for determination was available in Yang et al. [44].

### 2.3. Miseq Sequencing and Bioinformatics

Soil DNA was extracted from 250 mg frozen soil samples using a PowerSoil DNA Isolation Kit. A ca. 340 bp of the 18S rDNA gene was amplified using a two-step PCR, and primer pairs GeoA-2 [46]/AML2 [47] and NS31 [48]/AMDGR [49] were used in the first and second PCR reaction, respectively. The primer NS31 was labeled with a unique 12 nt barcode at the 5′ end to discriminate different samples. The detail information about the PCR conditions and quality assessment can be found in Yang et al. [50]. The PCR products were purified, thoroughly mixed and sequenced at the Environmental Genome Platform of Chengdu Institute of Biology, Chinese Academy of Sciences. The raw sequence data have been deposited on the NCBI SRA database (accession No. PRJNA882223). 

Low-quality sequences and potential chimeras were removed using QIIME Pipeline Version 1.8.0 (Boulder, CO, USA) [51] before further analysis. The remaining sequences were clustered into different operational taxonomic units (OTUs) with 97% similarity level using USEARCH v8.0 [52]. The representative sequences of each OTU were blasted against the NCBI nt database to remove non-AM fungal OTUs. The number of reads per sample was normalized to the smallest sample size using the ‘normalized.shared’ command in Mothur [53]. Then a neighbor joining tree was constructed in MEGA v5 [54] to precisely identify these OTUs. The tree was visualized with iTOL [55].

### 2.4. Data Analysis

Mean weight diameter (MWD) was used to represent for soil aggregate stability and calculated as follows:$$\mathrm{MWD}=\sum_{i=1}^{n}\bar{Xi}\;\times \;Wi$$

Here, *Wi* is the percentage of the ith aggregate; *Xi* represents the average diameter of the aggregate. 

Two-way ANOVAs were used to detect the impacts of biochar addition, compost addition and their interaction on the proportion of large macroaggregate, little macroaggregate, microaggregate and MWD. AM fungal α-diversity indices including Shannon diversity and Pielou evenness indices were calculated in the “vegan” package. Three-way ANOVAs were used to examine the effects of biochar addition, compost addition, aggregate fraction and their interaction on soil physiochemical variables and AM fungal alpha-diversity indices. Differences among treatments were tested by a Tukey’s HSD post-hoc test at *p* < 0.05. Random forest analysis [56] was used to explore the main predictors of AM fungal alpha-diversity indices using the “randomForest” package [57]. 

We used PERMANOVA to detect the effect of compost addition, biochar addition and aggregate fraction on AM fungal community composition with 999 permutations. The AM fungal community compositions were ordinated using principal co-ordinates analysis (PCoA) based on the Bray–Curtis dissimilarity matrices in the “vegan” package. Mantel tests were conducted to examine the correlations between AM fungal communities and soil variables with the “ecodist” package [58]. A Venn diagram was used to detect unique and shared AM fungal OTUs among aggregate fractions in the “VennDiagram” package.

The whole AM fungal co-occurrence network was built based on all soil samples. AM fungal OTUs occurred in >50% communities were included in the network. Spearman’s correlation coefficients were calculated between OTUs using the “Psych” package. Then the correlations between OTUs with a Spearman’s coefficient <0.7 and a *p* value > 0.05 were eliminated [59]. The ecological cluster (module) in network was identified using “cluster_fast_greedy” function in “igraph” package. Furthermore, AM fungal network in each treatment was built using data from all aggregate fractions, and the criteria was the same as mentioned above. The topological parameters in each network including connectedness, average degree, average betweenness were calculated using “igraph” package.

## 3. Results

### 3.1. Soil Physiochemical Variables

All soil physiochemical variables were presented in Appendix A. Two-way ANOVA analysis revealed that soil AP was significantly affected by compost addition and biochar addition (Appendix A). Compared with CK, compost addition enhanced AP content, while biochar addition showed the opposite effect (Appendix A). Soil AK was significantly affected by biochar addition and aggregate fraction, but unaffected by compost addition (Appendix A). Soil AK content in treatment B and BC was much higher than that in treatment CK and C across all aggregate fractions (Appendix A). Both TN and TP were significantly different among aggregate fractions, with microaggregate harbored the highest TP but lowest TN content (Appendix A). Then the soil aggregate stability was evaluated using aggregate fractions and MWD. Generally, the little macroaggregate make up the largest proportion of soil aggregate, while microaggregate occupies only a small proportion across all samples (Figure 1A–C). The proportion of large macroaggregate was significantly influenced by biochar addition, while the proportion of small macroaggregate and microaggregate were unaffected by biochar or compost addition (Figure 1A–C). By calculating MWD, we revealed that compost addition with or without biochar addition greatly enhanced soil aggregate stability (Figure 1D).

### 3.2. AM Fungal Diversity

Totally, 66 AM fungal OTUs were annotated at 97% identity across all samples after normalization (5064 reads per sample). Among these OTUs, Glomeraceae was dominant (44 OTUs, relative abundance of 44.5%), followed by Paraglomeraceae (3 OTUs, 30.37%), Claroideoglomeraceae (8 OTUs, 21.23%), Gigasporaceae (2 OTUs, 2.48%), Diversisporaceae (5 OTUs, 1.34%), Ambisporaceae (3 OTUs, 0.08%) and Archaeosporaceae (1 OTU, <0.01%, Appendix A). 

AM fungal alpha diversity was evaluated using Shannon diversity and Pielou evenness indices. Three-way ANOVA analysis indicated that these indices were all significantly affected by biochar addition, marginally affected by aggregate fraction, but unaffected by compost addition (Appendix A). Specifically, AM fungal Shannon diversity and Pielou evenness indices were significantly higher in treatment B and BC than in CK treatment (Figure 2A,B). Random forest analysis indicated that soil AN, AP and TP were the most important predictors for AM fungal Shannon diversity. At the same time, AN was the only significant predictor for Pielou evenness index (Figure 2C,D).

### 3.3. AM Fungal Community Composition

PERMANOVA analysis indicated that AM fungal community composition was significantly affected by biochar addition (*r*^2^ = 0.08, *p* < 0.001), while unaffected by compost addition and aggregate fractions. This result was also supported by PCoA ordination, where AM fungal community compositions only slightly overlapped between treatment BC and C, as well as between treatment B and CK (Figure 3A). Mantel test revealed that AM fungal community composition was significantly impacted by soil AN, and marginally impacted by soil AP. As shown by Venn diagram, there were up to 51 AM fungal OTUs (78.46%) shared among three aggregate fractions (Figure 3C). The large macroaggregate harbored the largest number of unique AM fungal OTUs, while there was only one unique OTU in microaggregate and no unique OTU in little microaggregate (Figure 3C), respectively. 

At family level, the relative abundance of Diversisporaceae (*F* = 7.65, *p* = 0.009) and Paraglomeraceae (*F* = 6.34, *p* = 0.02) were significantly affected by biochar addition. The relative abundance of Paraglomeraceae was consistently lower in treatment B than in CK across all soil aggregate fractions, while the Diversisporaceae exhibited the opposite trend (Figure 3B). However, other AM fungal families were unaffected by compost or biochar addition.

### 3.4. AM Fungal Co-Occurrence Network

The whole AM fungal co-occurrence network was constructed based all samples, which contained 29 nodes and 55 edges (Figure 4A). A total of five modules of AM fungal OTUs strongly co-occur with each other in this network. Specifically, Module #1, Module #2 and Module #3 was dominated by Glomeraceae; Module #4 was solely consisted of Gigasporaceae; module #5 consisted of Paraglomeraceae and Claroideoglomeraceae (Figure 4B). We then only focused on the modules with more than five nodes. Two-way ANOVA analysis revealed that Modules #1 and #3 were unaffected by compost addition or biochar addition (Figure 4C,E). However, Module #2 was significantly impacted by biochar addition (*F* = 14.35, *p* < 0.001), where it exhibited a higher relative abundance in treatment B and BC than in C and CK (Figure 4D).

AM fungal co-occurrence networks for each treatment were built by combing all aggregate fractions. Treatment CK and C have more extensive AM fungal networks than treatment B and BC (Figure 5A–D), which were also supported by their higher node numbers (Figure 5E). The complexity and connectivity of AM fungal networks were much higher in treatment CK and C than in treatment B and BC (Figure 5A–D). This pattern was demonstrated by their network topological properties, i.e., the connectedness, average degree and betweenness were significantly higher in treatment CK and C than in treatment B and BC (Figure 5E). In addition, treatment B harbored the lowest proportion of positive links as compared with other treatments (Figure 5E). We also constructed AM fungal networks in each aggregate fraction. As illustrated in Appendix A, the AM fungal network in large and little macroaggregate fractions were more complex and larger than in microaggregate fraction.

## 4. Discussion

In the present study, 66 AM fungal OTUs were annotated at 97% identity totally. Similarly, our previous studies obtained 45 and 66 AM fungal OTUs using the same primers [28,33]. Although our results only represented a small AM fungal diversity, it was comparable to our previous studies in the black soil region. Black soil is regarded as one of the most fertile soil types and possess phosphorus and nitrogen availability. It was proposed that high nutrient availability would reduce AM fungal growth and diversity, since their host plants would invest less carbon to AM fungi under such condition than in nutrient-poor condition [33]. In support of our first hypothesis, biochar exhibited stronger legacy effect for AM fungi than compost. This was interpreted from the results that one-time biochar addition significantly enhanced AM fungal alpha-diversity and modified AM fungal community composition, while one-time compost addition showed little control. Biochar is more durable than any other types of organic fertilizers [60]. The beneficial effects of biochar on soil nutrients, crop productivity and microbial diversity still existed after several years’ application [19,61,62,63]. Although kinds of literature relating to the effect of biochar addition on AM fungal community is limited, some studies have provided evidence that biochar addition could improve AM fungal growth. For instance, a recent field study reported that biochar amendment increased AM fungal spore density and colonization in wheat roots [64]. The following mechanisms would be responsible for this beneficial effect. First, as revealed by random forest analysis, soil nutrient availability (e.g., AN) all contributed to AM fungal diversity. Therefore, the positive effect of biochar addition was possibly due to the modified soil nutrient status, which has been repeatedly demonstrated [33,41]. Alternatively, the enormous surface area and macropores (>200 nm) provide a special habitat for AM fungi [65,66,67,68,69,70], which protect them from predatory soil microarthropods [65,67].

For compost, we did not observe a legacy effect of compost addition on AM fungal community. The absence of compost effect was possibly due to the resilience of AM fungal community after compost addition. In a field study, the soil microbial communities were highly resilient to organic amendment and returned to its initial status after 1 year’s recovery [71]. On the other side, previous study has reported that one time compost applications with high rate (>500 t·ha^−1^) for soil restoration provide >10-year benefits [72]. However, the application rate of compost in the present study is 10 t ha^−1^, which is far less than the compost rate in the study mentioned above. Therefore, the undetected beneficial effect of compost was also possibly due to (1) the application rate being insufficient to induce a shift of AM fungal communities; (2) most of the compost was mineralized for two consecutive years. Even though, the current study indicate that soil aggregate stability was indeed greatly improved by one- time compost amendment. Soil aggregate stability is highly dependent on organic matter contents, as SOM could act as binders and induce soil mineral particles aggregating [73]. As a consequence, the application of compost generally improve soil aggregate stability [74,75], and this effect can be detectable in 2 years [76]. 

AM fungi can form complex networks in soils, facilitating connections and nutrient transport among plants [77]. Therefore, understanding the complex AM fungal networks can provide a new insight into AM fungal functions. In the current study, the AM fungal networks harbored extremely high positive links across all treatments, indicating cooperation was the main interaction type among AM fungi or they responded to environment fluctuations in the same direction [78]. Our findings also have implications for understanding how complex AM fungal interactions respond to agricultural practices. Biochar addition, dramatically simplified AM fungal co-occurrence network and decreased the network size, confirmed by the lower connectedness and average degree. We speculated that the reduction of AP in biochar added soils could serve as a mechanism for this pattern, which possibly restrains AM hyphal extension and thus reduce interactions among AM fungi. Biochar has the capacity to absorb phosphate ions and protect organic P from mineralization, leading to decreased P availability in biochar added soils [79,80]. This pattern has been evidenced by Warnock et al. [80], in which biochar addition hampered AM fungal hyphal density via decreased soil AP. On the other side, the toxic substances (e.g., VOCs) in biochar would inhibit AM hyphal activity and weaken their interactions in soil [81,82]. 

The AM fungal communities within aggregate fractions were poorly understood albeit their essential roles in aggregate formation and stability [83,84]. In contrast to our hypothesis 3, AM fungal community composition was independent of soil aggregate fractions, which was interpretated by the PERMANOVA analysis as well as the fact that only a small number of AM fungal OTUs were shared among aggregate fractions. Our finding was consistent with Liu et al. [43], in which AM fungal community did not differ among aggregate sizes in a subtropical grassland. Different aggregate fractions provide spatially heterogeneous habitats and generate new ecological niches for AM fungi [68,69,70], and consequently distinct AM fungal community across aggregates was expected. However, our result was not surprising when considering AM fungal dispersal capacity. AM fungi can penetrate or entangle all soil aggregate fractions through extensive hyphae [85], thus facilitating their dispersal. Therefore, the dispersal ability makes AM fungal community composition relatively homogeneous at the microscale in the current study.

## 5. Conclusions

In conclusion, our study investigated the legacy effect of biochar and compost addition on AM fungal community. Our results revealed that biochar addition exhibited a more substantial legacy effect for AM fungi than compost addition. Specifically, biochar addition significantly increased AM fungal alpha-diversity, altered AM fungal community composition and simplified AM fungal co-occurrence network. Unexpectedly, soil aggregate fraction was not a decisive factor in shaping AM fungal community composition. Our findings are of significance for understanding the response of AM fungi to agricultural practices, and thus may offer an effective way to improve soil fertility through the management of AM fungal community.

## Figures and Tables

**Figure 1 microorganisms-10-02137-f001:**
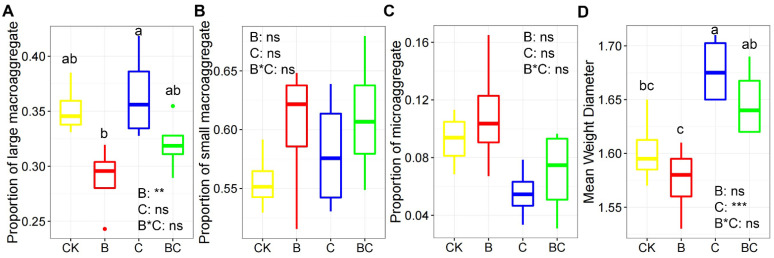
Box plot showing the proportion of large macroaggregate (**A**), small macroaggregate (**B**), microaggregate (**C**) and mean weight diameter (**D**) among treatments. Abbreviations: CK, control; B, biochar addition; C, compost addition; BC, biochar and compost addition. **, 0.001< *p* < 0.01; ***, *p* < 0.001; ns, not significant. The dot in the figure represents outlier. Bars without shared letters indicate significant difference at *p* < 0.05.

**Figure 2 microorganisms-10-02137-f002:**
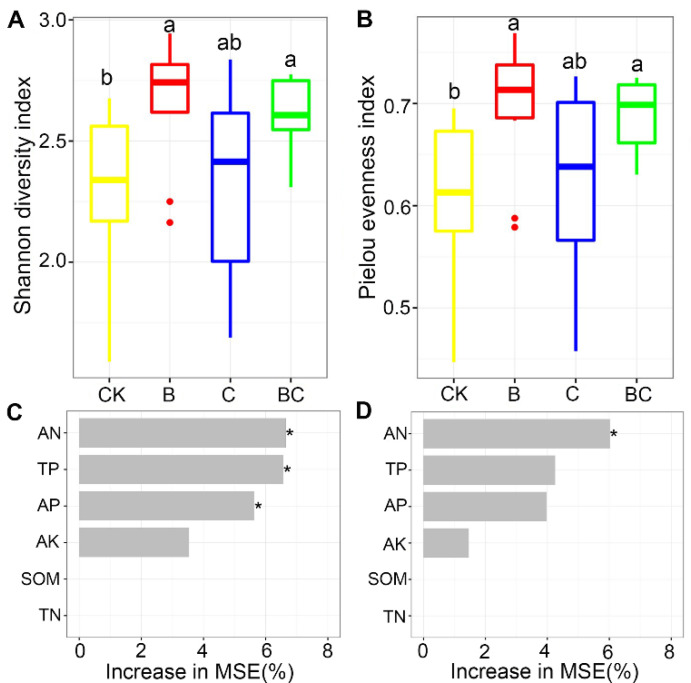
Arbuscular mycorrhizal (AM) fungal Shannon diversity indices (**A**) and Pielou evenness indices (**B**) among treatments; random forest mean predictor importance of soil variables on AM fungal Shannon diversity indices (**C**) and Pielou evenness indices (**D**). Abbreviations: CK, control; B, biochar addition; C, compost addition; SOM, soil organic matter; TN, total nitrogen; AN, available nitrogen; TP, total phosphorus; AP, available phosphorus; AK, available potassium; *, *p* < 0.05. The dot in the figure represents outlier. Bars without shared letters indicate significant difference at *p* < 0.05.

**Figure 3 microorganisms-10-02137-f003:**
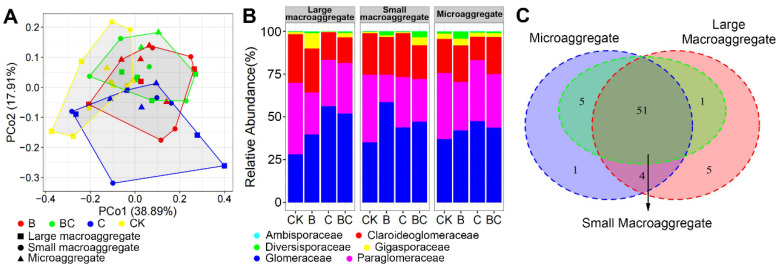
Principal coordinate analysis of arbuscular mycorrhizal (AM) fungal community compositions among treatments (**A**); relative abundance of AM fungal families among treatments in different aggregate fractions (**B**); Venn diagram showing the unique and shared AM fungal OTUs among different aggregate fractions (**C**). Abbreviations: CK, control; B, biochar addition; C, compost addition.

**Figure 4 microorganisms-10-02137-f004:**
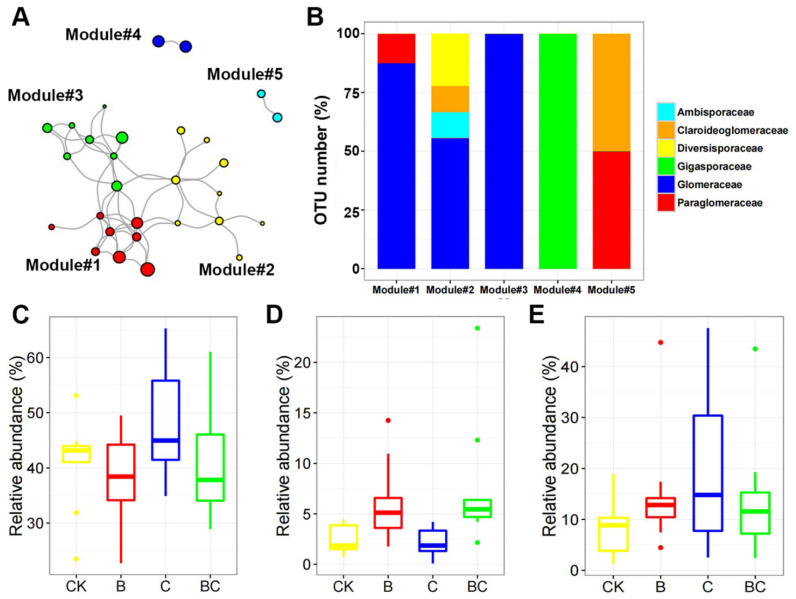
AM fungal co-occurrence network with nodes colored according to the modules (**A**); proportions of OTU number in the main AM fungal families within the five modules (**B**); relative abundance of Module #1 (**C**), Module #2 (**D**), Module #3 (**E**) among treatments. Abbreviations: CK, control; B, biochar addition; C, compost addition. The dot in the figure represents outlier.

**Figure 5 microorganisms-10-02137-f005:**
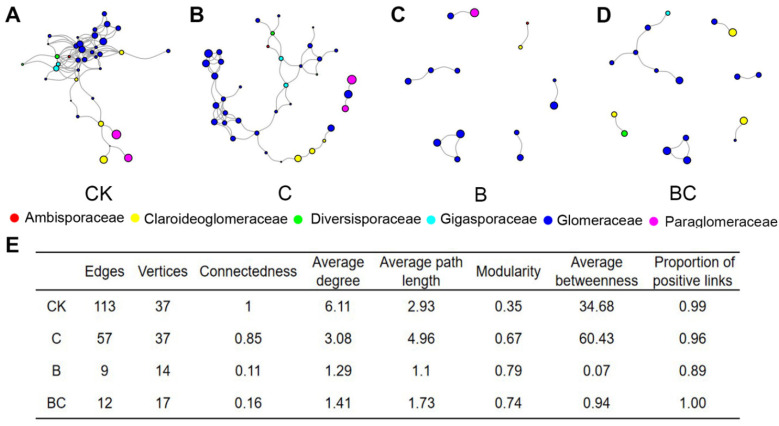
AM fungal co-occurrence networks in control (**A**), compost addition (**B**), biochar addition (**C**) and biochar with compost addition treatments (**D**); topological properties among AM fungal co-occurrence networks (**E**). In (**A**–**D**), the size of each node is proportional to its relative abundance, the color of each node represents family.

## Data Availability

Data are available by contacting the authors.

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
