# Peer review of "Legacy Effects of Biochar and Compost Addition on Arbuscular Mycorrhizal Fungal Community and Co-Occurrence Network in Black Soil"

_microorganisms, 2022, doi:10.3390/microorganisms10112137_

Round 1

Reviewer 1 Report

The manuscript submitted to me for review entitled “Legacy effects of biochar and compost addition on arbuscular mycorrhizal fungal community and co-occurrence network in black soil” is a research paper dealing with the application of biochar and compost derived from agricultural wastes to improve soil fertility, mycorrhizal development to increase food security.

The title is well structured and corresponds to the manuscript’s content. The abstract systematizes the research idea and main findings.

Generally, the manuscript sections are well developed. It is well-written and easy to understand. The abstract describes the most important findings and the general sense of the research. The key words are right-ones.

The introduction discusses all parts of the study. It is acceptable in the current estate.

The section Materials and methods is developed in detail and is positive from the point of view of research reproducibility. The analyses were performed using two- and three-way ANOVA and PERMANOVA. The authors used α-diversity indices, including Shannon diversity and Pielou evenness indices, principal coordinates analysis and different analytical packages to reveal the main dependences in the research.

Concerning results, they are well developed and in detail. The authors applied supplementary files also.

In my opinion, the discussion is well developed, comparing previous results with the authors’ one. The literature sources used in the manuscript are recent, as about 34% were published during the last five years.

The Conclusions are well supported by the results.

In my opinion, the manuscript could be accepted in the current estate. It will be well cited because the research is very actual. 

Author Response

Thank you for your comments. We really appreciate it!

Reviewer 2 Report

a work well set up methodologically and with strictly exposed and correctly analyzed results. Two small notes: L 60 "ways" no "wasy"; L 61 I suggest "soil volume" instead of "surface area"

Author Response

1. L 60 "ways" no "wasy"

Response: Revised in Line 60.

2. L 61 I suggest "soil volume" instead of "surface area"

Response: Revised in Line 61.

Reviewer 3 Report

The current study was conducted to investigate the long term effect of biochar and compost application on AMF community composition in high fertility black soil. Authors hypothesized that biochar has stronger effect on AMF community than that caused by compost. Also, this study indicated the long-term effect of one time addition of both biochar and compost. In addition to investigate the effect of soil aggregation on AMF community composition. Therefore, a field experiment was designed in blocks with treatments of with and without biochar and compost or in mix plus 3 soil aggregate fractionations. In general, the manuscript is scientifically well written and discussed an interested subject covering agronomical application of organic fertilizers on AMF community composition and networks.

Comments:

Why equivalent N content per h was only mentioned in treatments of C and BC? What about B treatment?

Does application rate of C and B fertilizers (10 t ha−1 ) equivalent to those applied for agronomical trail on that kind of fertilized black soils?

In section: 2.3. Miseq sequencing and bioinformatics. Using nested PCR increased the PCR bias. Why did one step PCR not use to amplify AMF community composition?

In section: 3.2. AM fungal diversity, L200-202

“Totally, 66 AM fungal OTUs were annotated at 97% identity across all samples after normalization (5,064 reads per sample).” Although AMF specific primers were used in this study, very small AMF diversity was represented. Authors should explain why AMF community composition was represented by very small diversity.  

Author Response

1. Why equivalent N content per h was only mentioned in treatments of C and BC? What about B treatment?

Response: We applied equivalent N content in treatments of C and BC, but not in B treatment. We adjusted N application amount in C and BC according to N content in compost. B treatment was not applied with compost, so we did apply equivalent N content in B treatment.

2. Does application rate of C and B fertilizers (10 t ha−1) equivalent to those applied for agronomical trail on that kind of fertilized black soils?

Response: Yes, the application rate of compost and biochar was equivalent to the recommended amount of fertilizer in this area. This information has been added in Line 103-104.

3. In section: 2.3. Miseq sequencing and bioinformatics. Using nested PCR increased the PCR bias. Why did one step PCR not use to amplify AMF community composition?

We used nested PCR because one-step PCR will amplify a larger proportion of non-AM fungi, protist and plants.

4. In section: 3.2. AM fungal diversity, L200-202

“Totally, 66 AM fungal OTUs were annotated at 97% identity across all samples after normalization (5,064 reads per sample).” Although AMF specific primers were used in this study, very small AMF diversity was represented. Authors should explain why AMF community composition was represented by very small diversity.

Response: In our previous studies (Yang et al., 2018; Yang et al., 2020), there were totally 45 and 66 AM fungal OTUs. Therefore the small AM fungal diversity was roughly equivalent to our previous studies in the black soil region.